# Grafting of Thiazole Derivative on Chitosan Magnetite Nanoparticles for Cadmium Removal—Application for Groundwater Treatment

**DOI:** 10.3390/polym14061240

**Published:** 2022-03-18

**Authors:** Mohammed F. Hamza, Adel A.-H. Abdel-Rahman, Alyaa S. Negm, Doaa M. Hamad, Mahmoud S. Khalafalla, Amr Fouda, Yuezhou Wei, Hamada H. Amer, Saad H. Alotaibi, Adel E.-S. Goda

**Affiliations:** 1School of Nuclear Science and Technology, University of South China, Hengyang 421001, China; 2Semi Pilot Plant Department, Nuclear Materials Authority, P.O. Box 530, El-Maadi, Cairo 11728, Egypt; mahmoudsayed24@yahoo.com; 3Chemistry Department, Faculty of Science, Menofia University, Shebin El-Kom 32511, Egypt; adelnassar63@yahoo.com (A.A.-H.A.-R.); meenmido000@gmail.com (A.S.N.); dodyhamad95@gmail.com (D.M.H.); 4Botany and Microbiology Department, Faculty of Science, Al-Azhar University, Nasr City, Cairo 11884, Egypt; amr_fh83@azhar.edu.eg; 5School of Nuclear Science and Engineering, Shanghai Jiao Tong University, Shanghai 200240, China; 6Department of Chemistry, Turabah University College, Taif University, P.O. Box 11099, Taif 21944, Saudi Arabia; h.amer@tu.edu.sa (H.H.A.); s.alosaimi@tu.edu.sa (S.H.A.); 7Tanta Higher Institute of Engineering and Technology, Tanta 31739, Egypt; adelgoda1969@gmail.com

**Keywords:** magnetic chitosan nanoparticles, functionalization, uptake kinetics, sorption isotherms, water treatment, industrial area

## Abstract

The synthesis and developments of magnetic chitosan nanoparticles for high efficiency removal of the cadmium ions from aquatic medium are one of the most challenging techniques. Highly adsorptive composite (MCH-ATA) was produced by the reaction of chitosan with formaldehyde and amino thiazole derivative. The sorbent was characterized by FTIR, elemental analyses (EA), SEM-EDX, TEM analysis, TGA and titration (volumetric). The modified material includes high nitrogen and sulfur contents (i.e., 4.64 and 1.35 mmol g^−1^, respectively), compared to the pristine material (3.5 and 0 mmol g^−1^, respectively). The sorption was investigated for the removal of Cd(II) ions from synthetic (prepared) solution before being tested towards naturally contaminated groundwater in an industrial area. The functionalized sorbent shows a high loading capacity (1.78 mmol Cd g^−1^; 200 mg Cd g^−1^) compared to the pristine material (0.61 mmol Cd g^−1^; 68.57 mg Cd g^−1^), while removal of about 98% of Cd with capacity (6.4 mg Cd g^−1^) from polymetallic contaminated groundwater. The sorbent displays fast sorption kinetics compared to the non-modified composite (MCH); 30 min is sufficient for complete sorption for MCH-ATA, while 60–90 min for the MCH. PFORE fits sorption kinetics for both sorbents, whereas the Langmuir equation fits for MCH and Langmuir and Sips for MCH-ATA for sorption isotherms. The TEM analysis confirms the nano scale size, which limits the diffusion to intraparticle sorption properties. The 0.2 M HCl solution is a successful desorbing agent for the metal ions. The sorbent was applied for the removal of cadmium ions from the contaminated underground water and appears to be a promising process for metal decontamination and water treatment.

## 1. Introduction

The treatment of wastewater, industrial and/or mining effluents through the removal of contaminated or leachate metal ions is a crucial aspect with a critical prospective for many researchers. This is due to the increasing necessity for regulation from the government to protect the community [1]. The discharge of such effluents may cause serious health problems. On the other hand, the increasing demand for valuable and rare metals is imperative for industrial needs, making the recycling of such spent effluents (valorization in sub-products and recovery of spent materials) a vital demand. Furthermore, such studies are concerned with the decontamination of green life cycles and environmental needs, which are essential for the development of new processes for heavy metals recovery. The widely used methods for metal elimination are solvent extraction [2], precipitation [3], electrolytic techniques [4], membrane separation [5]. Most of these techniques are facing limitations in both economic and technical uses as well as deficiency in the low concentration effluents treatment, which still has a large variety of heavy ions.

Ion exchange and chelating resins are the most suitable tools for the recovery of metal ions from low concentrate solutions such as waste effluents and wastewater (either surface or underground water) [6,7,8,9]. This technique has received prominent attention in recent decades [10,11,12,13] with either a bio-sorbent (renewable resources) structure, a structure fabricated as nanoparticles and synthetic resins [14,15,16,17], or magnetic composite micro sorbents [18]. Natural biomaterials such as chitosan, alginate, algal biomass or agriculture waste bear reactive groups for further modification and are ready for grafting to other groups, which are designed as beads or fin particles (micro or nano scale) to enhance the sorption capacity and improve the sorption kinetics. The binding character in the bio sorbents is the same as in the synthetic resins (i.e., ion-exchange or chelation on the naturally existing groups or the grafted moieties).

Chitosan (deacetylated chitin) is one of the most abundant polysaccharides. It acquires great attention due to its natural hydrophilic properties throughout the presence of various hydroxyl and amine groups. Such materials are beneficial for improving the loading capacity for metal cations by ion exchange or chelating properties (through the lone pair of electrons on these groups) [19,20]. Additionally, it shows a new channel for grafting new functional groups (i.e., amine or heterocyclic). Beside improving the capacity, it is also for using in a broad range of pH (in either acidic or alkaline medium). The crude pristine polymer (as chitosan or alginate) is a poorly porous material and the resistance to intraparticle diffusion is the main controlled sorption mechanism, which makes it necessary to modify the polymer through hydrogel formation that gives possibility for expanding the structure [21,22] or designing as micro or nano scale particles for thin layer structure and improve properties of the mass transfer [23]. The modification opens a new trend for both selectivity and/or relative enhancement of the metal loading capacity for the prepared magnetic chitosan nanoparticles [24,25].

This work is described in successive stages; firstly, the synthesis of magnetite core particles by thermal precipitation to form nano particle size; secondly, the one-pot precipitation of dissolved chitosan with the prepared magnetite in the presence of epichlorohydrin (acts as a crosslinking agent with hydroxyl and amine moieties) with hydrothermal co-precipitation at strong alkaline media (pH around 10) [26,27,28], the free amino groups were attached with that in the thiazole heterocyclic through formaldehyde crosslinker to yield the final nanoscale functionalized chitosan particles. The synthesized composite was tested toward Cd (II) through varied pH factors, uptake kinetics, sorption isotherms, and the tendency of selective properties on polymetallic simulated nature solution that contains equimolar concentration for the studied the characterization of metal ions binding with the most associated elements in the applied industrial or mining effluents. The sorption desorption cycles were investigated for loss in the loading capacity and the chemical stability before application to the contaminated water in the mining and industrial area. The sorbent was completely characterized through different various tools for chemical and textural properties.

## 2. Materials and Methods

### 2.1. Materials

2-amino-4-thiazolacetic acid 95%, epichlorohydrin (EPI; 99%), chitosan (Medium molecular weight; and 75–85% of AD, acetylation degree), calcium chloride anhydrous >97%, cadmium chloride hydrate CdCl_2_.xH_2_O (99%), and ferric chloride hexahydrate FeCl_3_·6H_2_O (97%) were supplied from Sigma Aldrich (Merck KGa, Darmstadt, Germany). Aluminum chloride hexahydrate (AlCl_3_·6H_2_O), Magnesium chloride hexahydrate (MgCl_2_·6H_2_O), zinc chloride (ZnCl_2_), and lead sulfate (PbSO_4_) that used in the selectivity experiments were obtained from Guangdong Guanghua, Sci-Tech Co., Ltd. (Shenzhen, China).

### 2.2. Synthesis of Sorbent

#### 2.2.1. Synthesis of Magnetic Chitosan Nano-Particles (MG-CH)

Magnetite chitosan was prepared by hydrothermally co-precipitation method [29] using iron(II) sulfate hydrated (6.62 g) and iron(III) chloride (8.68 g), adjusting pH by sodium hydroxide solution (5 M) at 45 °C, after the dissolution of 4 g chitosan in 200 mL acidified water (with 6% (*w*/*w*) acetic acid solution). The pH of the solution was adjusted to a value of 10–10.4. After precipitation, the reaction was maintained for 1 h at 85–90 °C under vigorous agitation condition. The prepared magnetite chitosan nanoparticles were collected from the solution by magnetic control, then washed several times by water and acetone for the next step.

#### 2.2.2. Synthesis of Crosslinked Chitosan Nanoparticles (MCH)

This step enhances the chemical stability of the prepared nanoparticles through crosslinking effect and prevent their dissolution in acid solutions. The wetted magnetic chitosan nanoparticles from the previous step were mixed in an alkaline solution at pH 10 of 0.01 M EPI (0.067 M NaOH solution,) with a molar ratio of 1:1 of EPI solution and chitosan magnetite nanoparticles. The mixture was agitated for 2 h at 45 °C. The produced material (crosslinked chitosan) was collected by the magnetic bar and washed several times with water and ethanol to remove the unreacted materials.

#### 2.2.3. Synthesis of Grafted Amino Thiazol Acetic Derivative Nano Particles (MCH-ATA)

The mixture of activated chitosan nanoparticles (4 g) was added to a round bottom flask containing mixture of 2-amino-4-thiazolacetic acid (5 g) dissolved in 70 mL water, 10 mL of formaldehyde solution, and 1 mL of acetic acid (for adjusting pH to around 3). The mixture was refluxed for 7 h at 95 °C, the final product (MCH-ATA) was separated using magnetic control then rinsed with ethanol and water then air-dried at 50 °C for 5 h.

### 2.3. Characterization of Sorbents

The elemental analysis (EA) for sorbents (C, S, N, H and O composition) was investigated using the element analyzer (CHNOS, Vario EL III, elemental analyzer, Elementar Analyzer system GmbH, Sonaustraβe, Germany). The thermogravimetric (TGA) analysis of the nanocomposites was investigated using the TG-DTA (Netzsch, STA: 449 F3, Jupiter, NETZSCH-Gerätebau HGmbh, Selb, Germany); the analysis was operated under nitrogen atmosphere, with temperature ramp (10 °C min^−1^). The pH of zero-charge (pH_PZC_) was determined using the pH-drift method; a specific amount of sorbent (100 mg) was added to 50 mL of a solution containing 0.1 M NaCl with fixed initial pH (pH_0_) range from 1 to 11 values. After agitation for 48 h, the final pH (pH_f_) was determined and compared with the initial values (pH_0_). The pH_PZC_ is corresponding to pH_0_ = pH_f_. The FT-IR spectra (using KBr disk) were determined using IRTracer100, FT-IR spectrometer, with model Shimadzu-Tokyo-Japan. The analytical procedures were performed on the dry samples (at 60 °C) before grinding with KBr for conditioning as disk.

The amine content was estimated by volumetric titration [30,31] (30 mL (0.05 M HCl solution); C_HCl,1_ was mixed to 0.1 g of nanocomposite material with agitation for 20 h. The residual HCl concentration (C_HCl,2_) was determined by titration against 0.05 M NaOH solution, using phenolphthalein as pH indicator. The amine concentration was determined by the following equation:(−NH_2_) = (C_HCl,1_ − C_HCl,2_) × 30/0.1(1)

The morphological studies and chemical composition (qualitative) were measure using SEM-EDX analysis of samples. Scanning electron microscope (SEM), using Quanta FEG 200, (FEI France, Thermo Fisher Scientific, Merignac, France), which was coupled with the energy-dispersive X-ray diffraction analyzer (EDX analysis), Oxford Inca 350, EDX microanalyzer, Oxford Instruments France, Saclay, France. The initial and final (pH equivalent) of the solution were measured using a pH/ionometer, compact (S220 Seven), Mettler-Toledo China (Shanghai). The collected samples were filtrated by micromembrane (1.2 µm) before test.

### 2.4. Sorption Studies

The sorption studies were achieved using batch techniques. The prepared solutions (herein called synthetic solutions) were prepared through dilution of stock 1000 ppm metal solutions, the pH of these solutions was adjusted using 0.1/1 M HCl or NaOH solutions. The pH values were not controlled during the sorption processes, but it was monitored after the sorption (at the end) experiment. All experiments were triplicated for reproducibility evaluation and the figures have the average values of the three experiments with the error bar; the overall deviation did not exceed 4%, in which the standard deviation of these values was less than the size of symbols (if no error bars are much seen). The experiments (pH, kinetics, and isotherms) were performed through the mixing of a given amount of the sorbent (m, g), with initial concentration of metal ion (C_0_, mg L^−1^ or mmol L^−1^) and a fixed volume of the prepared solution (V, L) at certain pH value. The sorbent dosage was fixed for most experiments to 400 mg L^−1^_,_ except for the selectivity test that fixed to 1000 mg L^−1^. The pH of the solution was varied (1 to 6) for the pH investigation effect and the selectivity experiments, while other experiments of isotherms and kinetics were fixed at pH 4. The metal concentration was fixed to 100 mg Cd L^−1^, in which the sorption isotherms varied from 10–500 mg Cd L^−1^. The contact time of the experiments (pH and isotherms) was studied at 24 h for emphasizing complete sorption, while in the kinetic experiments, the sorption time was varied from 2 min to 48 h. After sorption, the samples were filtered using membrane; 1 µm pore size and the residual (C_eq_, mg L^−1^ or mmol L^−1^) of metal concentrations were measured using the inductively coupled plasma atomic emission spectrometer; ICP-AES (Activa M, Horiba France, Longjumeau, France). The capacity of sorption (q; mg g^−1^ or, mmol g^−1^) was measured using the mass balance equation q = (C_0_ − C_eq_)V/m. The desorption experiments and recycling were estimated using HCl solution with conc. 0.2 M. It was noticed that the desorbing rate being faster than the sorption one. In the recycling experiments, a rinsing step was performed systematically after each elution.

### 2.5. Uptake Kinetics and Sorption Isotherms Models

The uptake kinetics was modeled using PFORE (pseudo-first order rate equation) PSORE (pseudo-second order rate) and RIDE (Resistance to intraparticle diffusion) equations, while the isotherms studies were investigated by the Langmuir, Freundlich, Sips (the so-called Langmuir–Freundlich model) and Temkin equations. These models were reported in the Appendix A.

### 2.6. Treatment of Real Metal-Containing Groundwater

The sample of contaminated water was collected from Tura town, Egypt. The area is closed to the Nile River (Appendix A), and it is situated about half distance between Cairo and Helwan. Tura was known for quarrying activities by the ancient Egyptians as it was used for the mining of limestone and marble. Nowadays, the town is famous for many industrial activities besides quarries, including the cement industry. The mining industries significantly increase air pollution in the area not to mention soil contaminated [32], which explains the contaminated groundwater by the heavy elements such as Hg, Cd, Pb, Al, and Fe (Hazardous ions of interest). Appendix A shows the maximum contaminated levels for either drinking water (D.W) or livestock drinking water (L.W). Sorption was investigated at different pH values: 2.0, 4.0, 6.2 (the original pH) and 7.0, with the contact time of 10 h with sorbent dosage 5 g L^−1^. The variation of pH is occurred to shows its effect in the sorption behaviors in such diluted polymetallic solutions. After sorption, the filtrated samples were analyzed for residual concentration to calculate removal efficiency and loading capacity. The other way compares the residual concentrations on that of the maximum concentration levels (MCL) for either drinking water (D.W) or livestock drinking water (L.W).

## 3. Results and Discussion

### 3.1. Sorbent Characterization

#### 3.1.1. Morphology and Textural Properties

Figure 1 shows the TEM and SEM photographs of the synthesized modified sorbent. Five to ten micron is the overall average particle size with irregular shapes was obtained from SEM analysis Figure 1a. The TEM analysis confirms embedment of the nanoparticles (the dark points are the magnetic nanoparticles), which confirms that it is not uniform: as round particle predominates. The magnetite nanoparticles range are around 2–5 nm with homogeneously spreading in the polymer matrix as shown in Figure 1b.

Appendix A reports the S_BET_ (specific surface area) and the V_p_ (porous volume) of the sorbents. The functionalization process affects the textural properties, in which the S_BET_ varied from 21.17 to 22.5 for MCH and MCH-ATA, respectively, while the V_p_ values are around 7.14 to 7.7 cm^3^ STP g^−1^, respectively.

#### 3.1.2. Thermogravimetric Analysis

The thermogravimetric analysis shows symmetric degradation profiles of MCH and MCH-ATA sorbents as shown in Figure 2. The first step of degradation concerns with loss of absorbed water, this step is the same for both sorbents with approximately values 10.261% and 9.938% for MCH and MCH-ATA, respectively, the temperature profiles of these losses are around 237.8 and 246.5 °C for both sorbents, respectively. The second loss stage is concerned with the depolymerization of chitosan backbone and biopolymer. The loss in magnetite chitosan (MCH) is greater than functionalized one (around 27.01 and 23.764%, respectively) at temperatures of 352.1 and 419.6 °C, respectively. The final stage of loss is related to the decomposition of the polymer backbone and the complete degradation of the polymer skeleton. This loss stage for the MCH-ATA is much higher than MCH and found to be 39.864% and 9.69%, respectively. The final loss of both sorbents was recorded around 46.96% and 73.566%, respectively, which verifies successive functionalization reactions, and that the residue in the MCH is theoretically and practically greater than the functionalized sorbent. The DrTG, as shown in Appendix A, confirms close profiles with different stages at three extrema assigned as follows:(a)75.97 °C, 254.8 °C and 370.3 °C for MCH;(b)74.4 °C, 282.6 °C, and 471.5 °C for the MCH-ATA.

The functionalized sorbent shows extrema at higher temperature with relevant peaks and less enlarged than MCH.

#### 3.1.3. FTIR Spectroscopy of Synthesized Sorbents

The FTIR spectroscopy for the MCH, MCH-ATA, after metal sorption and after 5 cycles of sorption desorption is shown in Figure 3 and Table 1. This allows us to verify the modification and chemical changes during sorption properties. The -NH and -OH stretching vibrations overlapped peaks were noticed at 3400–3450 cm^−1^. The C = O stretching with the C = C of heterocyclic ring and C = N of amide group appeared at 1638 cm^−1^ for MCH, and 1620 cm^−1^, 1621 cm^−1^ and 1619 cm^−1^ for MCH-ATA, after sorption and after elution, respectively, while the amine bending vibrational peaks were shown at 1513 cm^−1^ and 1511 cm^−1^, respectively. The broadness of C = O peaks for the MCH-ATA after 5 cycles of elution was overlapping this peak. The β-D-glucose, which is related to chitosan moieties, was found in a series of peaks in the range 1150–850 cm^−1^. The magnetite particles Fe-O were confirmed through a peak at 572 cm^−1^ for MCH and 633 cm^−1^ for MCH-ATA [33]. The main noticed points for loaded sorbents are related to decreasing the intensity of peaks more than shifts, these functional groups were included in the binding mechanisms (as OH, NH, COOH, and S groups), and they are discussed briefly in the items below:(a)Decreasing the intensity peaks assigned to OH and NH with shifts from 3447 cm^−1^ and 3197 cm^−1^ to 3404 cm^−1^ related to sharing in the binding with metal ions;(b)Strong decreasing in the C = O stretching and amine binding vibration;(c)Decreasing of the COO^−^ salt, and –OH bending.

This is evidence for involving these peaks (-NH, –OH, and -C = O or -COO^−^) in a binding mechanism.

#### 3.1.4. Elemental Analysis

Appendix A reports the elemental analysis of the two sorbents before and after modification. These sorbents show nitrogen contents of 4.28% (3.056 mmol g^−1^) and 6.09% (4.348 mmol g^−1^) for MCH and MCH-ATA, respectively, which give evidence for modifications by increasing the amine contents through grafting of the dithiazole moiety. Oxygen contents also increased through grafting from 31.18% (19.762 mmol g^−1^) to 35.94% (22.464 mmol g^−1^) from the carboxylic groups in the grafted moieties.

The EDX analysis from Figure 4 (semi-quantitative analysis) shows increasing in the percent of O and N, which parallel to the elemental analysis. The appearance of S (Figure 4 for MCH-ATA) in the EDX chart confirms the successful grafting of thiazole moiety.

#### 3.1.5. Surface Charge Analysis—pH_PZC_

From the synthesis procedure, as shown in Figure 1, it was expected to observe different tautomeric structures of the thiazole ring (Figure 2). These forms are affected by the acid–base properties of MCH-ATA. Figure 5 compares the pH_pzc_ profiles of MCH and MCH-ATA using the pH-drift method. The modification is reflected by the shifts of pH_PZC_ from 6.23 for MCH to 5.19 for MCH-ATA. The MCH-ATA sorbent bearing acidic carboxylic groups which are the main reason for acidic shifts, in which the pK_a_ of the 2-amino-5-thiazole acetic acid (ATA) is around 3.2 (2.5 for thiazole).

### 3.2. Sorption Properties

#### 3.2.1. Effect of pH

The pH study is not only affected by the dissociation performed for the reactive groups at the sorbent surface but also for the metal speciation (including precipitation) in the solution. Appendix A shows the dissociation diagram of Cd(II) at the experimental sorption conditions.

Cationic cadmium species (CdCl^+^ and Cd^2+^) are largely predominated at the higher pH values. At pH higher than 3, the most cationic species that appears is Cd^2+^. Figure 6 shows the effect of the pH on Cd(II) sorption using MCH-ATA sorbent and comprises the capacity with the unmodified one (MCH). The experiments were repeated three times for the reproducibility evaluation and the figures contain the average values with the error bars, which indicate a good reproducibility of the sorbent for Cd sorption (the deviation from the three experiments is less than 4%). As expected, the increase in the pH values improves the Cd sorption efficiency, which is due to a progressive increase in the cationic individual species of Cd(II) metal ion and the deprotonation on the functional groups (O and N in MCH and O, N and S in the MCH-ATA) that decreases the repulsion effect toward cationic species and makes the electron pair on the chelating groups more available for binding. It is noteworthy that as the temperature of the medium increases, the sorption capacity is gradually increased, which is due to an increase in the swelling of the sorbent network that assists the diffusion of metal ions inside the pores as well as increasing the metal movement for easy bonding by the function groups.

From the pH sorption diagram, it is noticed that enhancing the sorption performances for both sorbents, as pH increased up to 4, the performance was almost stable for the MCH-ATA, while it was slightly increasing for the MCH sorbent with for a further increase in the pH.

The maximum average (from the triplicated experiments) sorption capacity for MCH is around 0.37 mmol Cd g^−1^, while it increased to 1.1 mmol Cd g^−1^ after functionalization at 22 (±2) °C and 1.4 mmol Cd g^−1^ at 50 °C.

Appendix A (see AMS) shows the variation in the pH values during Cd(II) sorption for both sorbents. In most cases, a slight decrease in the pH was noticed for MCH-ATA and MCH (Δ pH around 0.4 unit).

The intersecting pH_eq_ range is 3–5, which is corresponding to the optimum pH (Appendix A). The slopes of acidic media (corresponding to +0.46, +0.51, and +0.57) for MCH, MCH-ATA and MCH-ATA at 50 °C, respectively. The slope of this curves means uses of two protons in the ion exchange process per metal ion, while the partial negative charge on nitrogen and oxygen as well as that on the carboxylate ions are responsible for neutralizing the positively charged cadmium ions. The linear sections are identified as follows: (a) linear increasing of log K_d_ with the pH of the solution, followed by (b) the stabilization region.

#### 3.2.2. Uptake Kinetics

The kinetics profiles are reported in Figure 7, where it is shown a fast sorption property of the functionalized sorbent MCH-ATA compared to the non-functionalized one (MCH), under the selected experimental condition (C_0_: 100 (±5) mg L^−1^; SD: 200 (±5) mg L^−1^, pH_0_: 4 (±0.05)): fast sorption kinetics for functionalized sorbent (around 20–25 min) compared to the non-functionalized one (around 60–90 min). The sorbent size is the main factor for decreasing the sorption time, which explains the mass transfer properties. The thin layer of the coated polymer on the magnetite particles (as shown from TEM and SEM analysis), causes a reduction in the effect of intraparticle diffusion properties of metal ions. The reaction rate of kinetic properties was controlled by film, bulk and intraparticle diffusion rate. The resistance to diffusion is neglected by the fast sorption properties. The preliminary tests showed that maintaining the agitation velocity around 200 (±10) rpm avoids the sorbent sedimentation and minimizes the effect of resistance to bulk and film diffusion. By comparison, the determination of coefficients values, the equilibrium sorption capacities (experimental vs. calculated values), and the AIC values of the three models, confirms that the pseudo-first order rate equation (PFORE) better fits the kinetic profiles for both sorbents (MCH and MCH-ATA) than the PSORE and the RIDE. The interesting results were obtained by Hubbe [43] and Simonin [44] regarding the interpretation of the sorption modeling. In most cases, the variation in the concentration of metal ions in the solution affected the sorption mechanism. The selection of the appropriate experimental conditions can be controlled and can also predict the kinetic profile mechanisms (chemical or physical sorption). The uncontrolled design of the experiments may orient the reaction mechanism to the PSORE, whereas the system is controlled by the resistance to intraparticle diffusion.

Figure 7 shows the PFORE (average values of the three repeated experiments and the error bars of the results), while Table 2 shows the parameters for the two profiles. This also reports the correlation coefficients parameters for PFORE and PSORE, which indicate the calculated and experimental values, that sorption capacities (calculated) are closer in the PFORE than that in PSORE. Appendix A shows the PSORE of both sorbents, which does not fit the experimental condition.

#### 3.2.3. Sorption Isotherms

Sorption isotherms were triplicated (while the average values were plotted with error bars as shown in Figure 8) to confirm the reproducibility of sorption performances; this was performed at room temperature. The three-time repetition of experiments was typical in the profiles with a limited deviation. The cadmium sorption is characterized by steep initial slope before the saturation plateau as in Figure 8. Table 3 shows the determination coefficients (R^2^), which is the significant parameter for evaluation, in which MCH fits the experimental data with the Langmuir equation, while the MCH-ATA fits with the Langmuir and Sips equations (the parameters of the three profiles Langmuir, Freundlich and Sips were summarized in Appendix A). Appendix A displays the comparison of non-fitted equations of Freundlich and Temkin for MCH and MCH-ATA.

The Freundlich equation is used to describe a power-type equation, which is not fitted in our case. While the Langmuir model (assumed the physical adsorption) achieved through monolayer adsorption of the homogeneous system without interactions between the sorbed molecules with a homogeneous distribution of sorption sites on the sorbent surface.

NH, OH, COOH and S are the functional groups responsible for sorption of metal ions, through rearrangements and the tautomerization effect. The pH of the solution makes a partial deprotonation of the sorbent, which causes more availability of electrons on the groups (especially NH and COOH) for chelation.

According to the Pearson’s rules (HSAB) [45], Cd(II) is classified as a soft acid, which is favored over soft base. The N- and S-containing ligands are considered soft ligands and have high affinity for such metal, while O-containing ligand favors hard acids. These ligands are partially protonated and tautomerization rearrangements that explain the unexpected trend. Table 3 shows a list of sorbents toward the sorption properties of Cd(II). Some sorbents appeared with higher sorption for Cd metal ion. Besides the high sorption properties of this sorbent compared with the others, the sorption kinetics is the main advantage that the complete adsorption was performed in less than 30 min.

Table 4 compares the sorption capacity, equilibrium time, pH of the solution, and temperature of the sorption condition for the synthesized sorbent for Cd(II) with other sorbents. However, this comparison is not easily performed because of the unsystematic approach of experimental condition. But it seems to be promising uses because of high loading capacity comparing to most of them as well as short saturation time.

### 3.3. Selectivity–Sorption in Multi-Metal Solutions

The investigation of sorption selectivity at varied pH values (2–6) with equimolar major elements (Ca, Mg, and Al) found in mining ores such as gibbsite and shale in and some heavy metals that are present in contaminated water or industrial effluents (Fe, Zn and Pb) also found in some ore materials as ferruginous sandstone and gibbsite ore materials. The K_d_ (distribution coefficient) is determined through mass balance equation (K_d,Me_ = q_eq_/C_eq_, L g^−1^) as well as the selectivity coefficient SC_(Me1/Me2)_ = K_d,Me1_/K_d,Me2_. Figure 9 shows the average data of duplicated experiments at different pH values.

Surprisingly, the sorbent preference Cd and Pb over other elements at higher pH values in multi-metal solutions. The SC_(Cd/Ca, Mg or Al)_ is much higher than other elements and follow the same profile, in which the SC increases with the pH. In the acidic medium, the SC of these elements were not more than 10 times higher at pH_eq_ 3.27 and 3.18, while it increased by 12.1, 15.4 and 16.6 times for Ca(II). The same trend was observed for Mg(II), which increased by 8.5, 12.9 and 13 times, the selectivity of Al(III), which increased by 9.8, 14.9 and 15.2 times for pH_eq_ 4.28, 5.18 and 6.69, respectively. The SC_(Cd/Fe and Zn)_ shows lower values than Ca(II), Al(III) and Mg(II), which indicates a preference of sorbent to these elements. The selectivity of SC_Cd/Fe_ is around 5.8, 9.6, 10.7, 7.3 and 7.6 times higher at pH values 2.47, 3.18, 4.28, 5.18 and 5.69, respectively, while for Zn it was 4.0, 7,71, 7,07, 6.48, and 6.59 times, respectively. Less selectivity was recorded for pb compared to other elements, especially at high pH values, being 2.22, 2.83, 5.74, 6.56 and 6.19 times higher, respectively. This indicates the high affinity of the sorbent toward Cd(II) over heavy elements, but it not actually used for real separation individually from the solution. It can be used for the desalination of contaminated water and real effluents.

### 3.4. Metal Desorption and Sorbent Recycling

Desorption of adsorbed metal ions is an important criterion for evaluating the sorbent efficiency. This not only controls the valorization and recovery of the target metals but also for recycling the sorbents for further uses. The desorption process was performed by using a solution of 0.2 M HCl solution. The sorbent shows chemical stability after five cycles of sorption desorption as well as the stability of the magnetite core [58]. This is observed from the EDX analysis reported in Appendix A, which contains Fe with the same ratio as in the pristine sorbent, and the high level of Cl ions is derived from the eluent used (0.2 M HCl). From previous work [56,59], the loss in iron (magnetite core) did not exceed 1.1% after five cycles using high concentrate acid 0.5–1 M HCl, so loss with 0.2 M HCl seems to be neglected. The loss in the capacity of the sorbents shows a limited decrease in sorption performances; for MCH, it decreased from 16.15 to 14.52 mg g^−1^, which represents a decrease of 10% compared to 3.7% for the functionalized sorbent, where the capacity decreased from 46.89 to 45.17 mg g^−1^, as reported in Table 5. This indicates the stability of the final sorbent chemically and physically (not degradable), in keeping with the same capacity as the first use. It was emphasized by the FTIR analysis (see Figure 1) after cycles of sorption desorption, which appeared to restore the most interested functional groups (C = O, COO^−^, COOH and NH_2_) with as many peak intensities as possible. The elution efficiency of both sorbents stays around 100%, which indicates other trend properties for using these composites as well as the chemical stability in elution properties.

Appendix A shows a comparison study of desorption kinetics of the loaded sorbent (MCH-ATA), which indicates efficient uses of 0.2 M HCl for this purpose, and the contact time is much smaller than the sorption kinetics. Fifteen minutes was sufficient for complete desorption of sorbed cadmium ions. Appendix A displays the typical profiles of the triplicated experiments for both sorbents that are required for reproducibility.

### 3.5. Treatment of Contaminated Water

The functionalized sorbent was tested in the sorption technology for groundwater treatment. Appendix A reports the most contaminated elements found in the sample with a comparison to the MCL values of drinking water and livestock water.

The sorption was performed at different pH values (2, 4, 6.1 (original pH) and 7). As reported, in the sorption performances from individual elements, the sorption efficiency increased with pH. Under selected experimental conditions, the sorption efficiencies were higher than 90% for Cd(II) and Pb(II) at neutral pH value. While the sorption of Zn(II) is around 67%, the sorption performance at pH 7 and 6.2 is higher than at 4 and 2. The residual concentrations for the heavy elements compared to the maximum levels for drinking water (MCL, maximum level of concentration) with respect to the World Health Organization [1]. Figure 10 displays the removal efficiency of each element at different pH values, which emphasizes the increase in the removal efficiency with pH, while Appendix A shows the comparison of the sorbent with respect to drinking water and livestock water. In the case of livestock drinking water levels, it was fitted to the MCL [60] at high pH values and a little higher than drinking water for Cd(II) (around 1.3 times) and Hg(II) (around 2 times), initially at pH 7. It means that the sorbent can be used for the removal of the toxic elements from the high pH value at the normal pH of the water (around 6.2).

## 4. Conclusions

A highly adsorptive composite was prepared based on magnetite nanoparticles after grafting of thiazole moieties on chitosan particles by the effect of formaldehyde as crosslinking reagent. The sorbent was crosslinking for the chemical enhance stability. It is full characterized through FTIR, TGA, SEM, SEM-EDX, elemental analysis (EA), and titration. The PFORE is the most fitted for uptake kinetic, in which a limitation of the intra particle diffusion is revealed to the small particle size and the thin layer of organic matrices on the magnetite core materials. Langmuir equation fits sorption isotherms for MCH, while Langmuir and Sips for MCH-ATA sorbent. The loading capacity was improved by functionalization, it was increased from 0.61 mmol Cd g^−1^ for MCH to 1.78 mmol Cd g^−1^ for MCH-ATA. The functionalized sorbent shows a high stability through sorption recycles for at least 5 cycles. The sorbent was tested on a real solution supplied from an industrial and mining area, this solution was highly contaminated with toxic and heavy metals. The sorbent shows a good ability for cleaning this solution, fits with the livestock drinking water levels, and is a little high for drinking water on Cd (around 1.3 times) and Hg (around 2 times) at pH around 7, according to the WHO. The sorbent is a good tool for the removal of heavy metals from highly contaminated ground water.

## Data Availability

Data available from authors.

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
