# Peer review of "Grafting of Thiazole Derivative on Chitosan Magnetite Nanoparticles for Cadmium Removal—Application for Groundwater Treatment"

_polymers, 2022, doi:10.3390/polym14061240_

Round 1

Reviewer 1 Report

I have carefully read the manuscript MDPI Polymers-1610374 and find the work relevant, fitting the scope of the journal and likely to be of interest to a broad readership. The manuscript, however, needs some major revisions to be considered as publishable in the journal MDPI Polymers. In order to improve the quality of the manuscript, authors may take the following recommendations and comments into consideration:

  1. English needs to be polished and grammar improved considerably throughout the whole manuscript. The manuscript does not have an adequate academic level of English in its current form and needs to be improved significantly before being considered for publication.
  2. Abstract should be more specific and mention what was the achieved Cd load of the adsorbent (e.g. in mg/g) in the synthetic and the real groundwater.
  3. Lines 56-58: Authors state that “This technique received the attention through the last decades with either bio-sorbent (renewable resources) structure or fabricate as nanoparticles and synthetic resins”. In the context of this paper, I would add: “or magnetic composite microsorbents” and support this statement with the following literature reference: Drenkova-Tuhtan, A. et al. (2021). Sorption of recalcitrant phosphonates in reverse osmosis concentrates and wastewater effluents – influence of metal ions. Water Science & Technology (2021), 83 (4), 934-947. DOI: https://doi.org/10.2166/wst.2021.026.
  4. The synthesis steps of the Chitosan Magnetite Nanoparticles are supposedly based on already published procedures. Thus, sections 2.2.1, 2.2.2 and 2.2.3 need literature references on which the synthesis procedures are based.
  5. In section 2.2.1 the magnetic chitosan is abbreviated as MG-CH and in section 2.2.3 the grafted version of the same is abbreviated as MCH-ATA. Please, be consistent and use either MG-CH or MCH throughout the whole manuscript.
  6. Figure 6: The authors performed the pH experiments 3 times to confirm the reproducibility of the results and they plotted each one of those experiments individually, i.e. there are 3 curves for each dataset. This, however, is not necessary. It would be better if they plot one single curve for each material based on the average values from the 3 datasets and add the error bars showing the standard deviation for each data point.
  7. Figure 6: What was the initial concentration of Cd(II) in the pH experiments? Was it 100 mg/L? If yes, why so high? Such high concentration does not seem to be representative for polluted groundwater in real life.
  8. Figure 7: Same comment as Figure 6, i.e. merge the 3 curves from the 3 repetitions of each experiment into one single curve and add the error bars showing the standard deviation for each data point.
  9. Also, Figure 7 has bad aesthetics. No need to plot the data by including the full time series up to 48 h. It is obvious that the kinetics is fast and reaches equilibrium already after a couple of hours, so authors can cut the plot at 10 h (or 24 h) to magnify the time scale and be able to show closely only the first few hours where the actual kinetic process and drop in concentration take place. Moreover, the two separate graphs (left: MCH and right: MCH-ATA) can be combined in one after the 3 separate curves for each sorbent are merged into one. Last but not least, the font size of the graph text is too small and I recommend that the authors increase it a bit for a better legibility.
  10. Table 2: I suppose the middle line dataset should be “PSORE” and not “PFORE”.
  11. The scientific discussion of section 3.2.2. “Uptake kinetics” can be extended by analyzing and explaining what does it mean in a physical sense that the pseudo-first-order kinetics model fits better the experimental data.
  12. Figure 8: Same comment as Figure 6 and 7, i.e. merge the 3 curves from the 3 repetitions of each experiment into one single curve and add the error bars showing the standard deviation for each data point. Also, the font size of the graph text is too small and can be increased a bit for a better legibility. Moreover, the left and the right graphs for each model can be merged into one single graph.
  13. The scientific discussion of section 3.2.3 can also be extended by analyzing why the Langmuir model is the best fit for the experimental data and what does this imply.
  14. Table 4 can be improved by adding more information to facilitate an easier comparison between the studies, i.e. more columns can be added which include information about the initial Cd concentration, the sorbent dose / sorbent concentration, the temperature, the test scale, etc.
  15. Lines 411-412: “Figure 8” should be “Figure 9” and Lines 476-477: “Figure 9” should be “Figure 10”.
  16. Regarding the mechanical and chemical stability of the composite sorbent material, authors performed an EDX analysis of the reused sorbent material, but did they detect any leaching of the precursor elements / building blocks of the sorbent by measuring the concentration of those elements in the treated effluent?
  17. “Conclusions” section is repeated twice. Please, correct.
  18. The manuscript includes too many self-citations (at least 13 total), especially by Hamza, M.F. Please, reduce!

Author Response

Response to Reviewer #1

Red: Response to the Academic Editor’s comments

Blue: New information/comment added to the manuscript

First, we would like to thank you for your carful revision, evaluation and your valuable comments. We try best answering the comments and made the corrections, we hope it will meet with your approval.

 Open Review

English language and style

(x) Extensive editing of English language and style required
( ) Moderate English changes required
( ) English language and style are fine/minor spell check required
( ) I don't feel qualified to judge about the English language and style

Yes

Can be improved

Must be improved

Not applicable

Does the introduction provide sufficient background and include all relevant references?

( )

(x)

( )

( )

Is the research design appropriate?

(x)

( )

( )

( )

Are the methods adequately described?

(x)

( )

( )

( )

Are the results clearly presented?

( )

(x)

( )

( )

Are the conclusions supported by the results?

(x)

( )

( )

( )

Comments and Suggestions for Authors

I have carefully read the manuscript MDPI Polymers-1610374 and find the work relevant, fitting the scope of the journal and likely to be of interest to a broad readership. The manuscript, however, needs some major revisions to be considered as publishable in the journal MDPI Polymers. In order to improve the quality of the manuscript, authors may take the following recommendations and comments into consideration:

We would like to thank the reviewer for the positive evaluation of the work. This is appreciated.

The manuscript was carefully checked.

  1. English needs to be polished and grammar improved considerably throughout the whole manuscript. The manuscript does not have an adequate academic level of English in its current form and needs to be improved significantly before being considered for publication.

Thanks, We carefully revised the manuscript and made corrections of, English Language, typing mistakes and try best for correction the grammatical errors. We hope the revised version is now more understandable by the readers.

  1. Abstract should be more specific and mention what was the achieved Cd load of the adsorbent (e.g. in mg/g) in the synthetic and the real groundwater.

Thanks for notification we add the loading capacity data on the abstract for the synthetic solution and ground water

  1. Lines 56-58: Authors state that “This technique received the attention through the last decades with either bio-sorbent (renewable resources) structure or fabricate as nanoparticles and synthetic resins”. In the context of this paper, I would add: “or magnetic composite microsorbents” and support this statement with the following literature reference: Drenkova-Tuhtan, A. et al. (2021). Sorption of recalcitrant phosphonates in reverse osmosis concentrates and wastewater effluents – influence of metal ions. Water Science & Technology (2021), 83 (4), 934-947. DOI: https://doi.org/10.2166/wst.2021.026.

Thanks, we add the reference.

  1. The synthesis steps of the Chitosan Magnetite Nanoparticles are supposedly based on already published procedures. Thus, sections 2.2.1, 2.2.2 and 2.2.3 need literature references on which the synthesis procedures are based.

Thanks for comment, actually the preparation of magnetite and crosslinking are previously prepared, and we add the literature references but other synthesis procedure are new

  1. In section 2.2.1 the magnetic chitosan is abbreviated as MG-CH and in section 2.2.3 the grafted version of the same is abbreviated as MCH-ATA. Please, be consistent and use either MG-CH or MCH throughout the whole manuscript.

Thanks for the notification, there are a difference between them. The MG-CH (magnetite chitosan without crosslinking material) after treatment with magnetite only, while the MCH is the magnetite crosslinking chitosan after treatment with EPI and that abbreviation what used in the whole manuscript for this material.

  1. Figure 6: The authors performed the pH experiments 3 times to confirm the reproducibility of the results and they plotted each one of those experiments individually, i.e. there are 3 curves for each dataset. This, however, is not necessary. It would be better if they plot one single curve for each material based on the average values from the 3 datasets and add the error bars showing the standard deviation for each data point.

Thanks for notification; actually, we prefer to add the data of the three experiments to show (as mentioned) the reproducibility, but as you think that make sense and space we plot the average values of the three experiments.

  1. Figure 6: What was the initial concentration of Cd(II) in the pH experiments? Was it 100 mg/L? If yes, why so high? Such high concentration does not seem to be representative for polluted groundwater in real life.

Thanks for comment, you is right, but we usually use 50-100 mg/L solution, to be sure that metal ion is sufficient for the complete sorption and saturated of the sorbent through occupation of the functional groups. With the experimental condition, it was shown that, the removal % for the 100 mg/L solution is around 21%, 53% and 77% for MCH, MCH-ATA:20C and MCH-ATA-50C so uses of concentration lower than 100 mg/L will make risk for complete the correct sorption.

  1. Figure 7: Same comment as Figure 6, i.e. merge the 3 curves from the 3 repetitions of each experiment into one single curve and add the error bars showing the standard deviation for each data point.

Thanks, it was modified

  1. Also, Figure 7 has bad aesthetics. No need to plot the data by including the full time series up to 48 h. It is obvious that the kinetics is fast and reaches equilibrium already after a couple of hours, so authors can cut the plot at 10 h (or 24 h) to magnify the time scale and be able to show closely only the first few hours where the actual kinetic process and drop in concentration take place.

 Moreover, the two separate graphs (left: MCH and right: MCH-ATA) can be combined in one after the 3 separate curves for each sorbent are merged into one. Last but not least, the font size of the graph text is too small and I recommend that the authors increase it a bit for a better legibility.

Thanks for precise comments it was modified, merge and adjust the plot curve to 10 h, also improve the resolution and increasing the text font of the figures

  1. Table 2: I suppose the middle line dataset should be “PSORE” and not “PFORE”.

Thanks for notification, it was corrected

  1. The scientific discussion of section 3.2.2. “Uptake kinetics” can be extended by analyzing and explaining what does it mean in a physical sense that the pseudo-first-order kinetics model fits better the experimental data.

Thanks, this section was added

The preliminary tests were showed that maintaining of the agitation velocity around 200 rpm, this avoids the sorbent sedimentation as well as minimizes the effect of resistance to bulk and film diffusion. By comparison the determination coefficients values, the equilibrium sorption capacities (experimental vs. calculated values), and the AIC parameters of the three models, it confirms that the pseudo-first order rate equation, (PFORE) is more fits the kinetic profile for both sorbents (MCH and MCH-ATA) than the PSORE and the RIDE. The interesting results were performed by Hubbe [47] and Simonin [48] about the interpretation of the sorption modeling. In most cases the variation of the concentration of metal ions in the solution affect in sorption mechanism. The selecting of the appropriate experimental conditions can be control and predict of the of kinetic profile mechanism (chemical or physical sorption). The uncontrolled design of the experiments may orient the reaction mechanism to the PSORE, whereas in the system is controlled by the resistance to intraparticle diffusion.

  1. Figure 8: Same comment as Figure 6 and 7, i.e. merge the 3 curves from the 3 repetitions of each experiment into one single curve and add the error bars showing the standard deviation for each data point. Also, the font size of the graph text is too small and can be increased a bit for a better legibility. Moreover, the left and the right graphs for each model can be merged into one single graph.

Thanks, it was corrected with increasing the font of the graph text and try best for improving the resolution.

  1. The scientific discussion of section 3.2.3 can also be extended by analyzing why the Langmuir model is the best fit for the experimental data and what does this imply.

As it was known that the Freundlich equation is being a power-type equation which is not appropriate in the description of our case. On the other hand, the Langmuir model assumed the sorption achieved through monolayer adsorption (physically), homogeneous sorption system without interactions between the sorbed molecules (this supposed a homogeneous sorption sites distribution on the sorbent surface). The Sips equation is a combined model between Langmuir and Freundlich, which offers third parameter (adjustable parameter) that in most cases improves mathematical experimental fit.

This information was added to the text

 The Freundlich equation was described for a power-type equation which is not fitted our case, while the Langmuir model (assumed the physical adsorption) achieved through monolayer adsorption of the homogeneous system without interactions between the sorbed molecules with a homogeneous distribution of sorption sites on the sorbent surface.

 Table 4 can be improved by adding more information to facilitate an easier comparison between the studies, i.e. more columns can be added which include information about the initial Cd concentration, the sorbent dose / sorbent concentration, the temperature, the test scale, etc.

Thanks for notification, we try best to support the data in the table. We add condition of sorption temperature and the initial Cd concentration, we think the more extra data will be loaded on the table. Hope it will be suitable now

  1. Lines 411-412: “Figure 8” should be “Figure 9” and Lines 476-477: “Figure 9” should be “Figure 10”.

Thank you very much for carful revision, it was corrected in the legends and in the text

  1. Regarding the mechanical and chemical stability of the composite sorbent material, authors performed an EDX analysis of the reused sorbent material, but did they detect any leaching of the precursor elements / building blocks of the sorbent by measuring the concentration of those elements in the treated effluent?

May this comment not clear for us, but we measure the concentration of the elements before and after sorption and also after elution to detect the sorbent efficiency. Also, there are a previous study for degradation of magnetite core material after series of sorption desorption experiments and found a very limited discharge of the magnetite core by around 1.5% after long term of recycles. The stability in the sorption and desorption with a very limit decreasing in the efficiencies rule out the degradation of the functional groups, also the FTIR analysis after long term of sorption desorption shows restoring the functional groups as in the original modified sorbent that reflect any types of degradations.   

  1. “Conclusions” section is repeated twice. Please, correct.

Thanks, it was corrected

  1. The manuscript includes too many self-citations (at least 13 total), especially by Hamza, M.F. Please, reduce!

If fact, all of the citation are related to the subject, we try to remove as much as possible but actually will make defect. Thanks for understanding

Reviewer 2 Report

Comments

The paper entitled “Grafting of Thiazole Derivative on Chitosan Magnetite Nano-1 particles for Cadmium Removal; Application for Groundwater 2 Treatment” was reviewed. It includes some new findings for preparation of materials and metal adsorption, but there are so many unclear points and typos, so it is not suitable to be published from Polymers.

Main points:

  1. The role of magnetite is not clear. Is it just for recovery of adsorbent? So, why did the authors carried out Fe adsorption by using this material?
  2. In abstract, the authors mentioned “titration”, but they should mention which kind of titration was carried out.
  3. The magnetite is located between two hydroxy groups from epichlorohydrin in the first part of Scheme 1. However, it was incorporated in the first step described in 2.2.1. before crosslinking.
  4. Is “HCO” in the second part of Scheme 1 formaldehyde?
  5. It is well-known that aldehyde reacts with amine to form imine. However, in the second part of Scheme 1, formaldehyde crosslinked amine from chitosan and amine from 2-amino-4-thiazoleacetic acid. Is it correct?
  6. Related to comment 4, it seems that 2-amino-4-thiazoleacetic acid was directly attached with amino group of chitosan to form ion complex. Carbonyl peak of MCH-ATA in Figure 3 appeared around 1620 cm-1, which can be identified as not COOH but COO-.
  7. Which is the metal adsorption site? Related to comment 5, the authors should prepare the figure of pH dependency for log D vs. pHeq from Figure 6. If the authors obtain the slope of 2 for Cd adsorption, the adsorption is dependent on pH. But if not, this curve was caused by hydrolysis of Cd2+ and deprotonation of amino groups from chitosan and 2-amino-4-thiazoleacetic acid, not from dissociated carboxylic acid.
  8. The authors described selectivity in 3.3. What was derived from? If it was derived from the attached 2-amino-4-thiazoleacetic acid, how about the stability constant of metal ions with 2-amino-4-thiazoleacetic acid? The adsorption reactions must be very complicated and the materials are not pure, so the discussion seems to be non-sense. Another lot adsorbent has another data.
  9. How about the attached amount of 2-amino-4-thiazoleacetic acid on chitosan? Is that related to the Cd adsorption capacity?
  10. The authors described two adsorption mechanism. What these mechanisms express for your adsorption?
  11. Why did authors describe tautomeric effect of thiazole ring? What this effect contribute to?
  12. Are the pH values mentioned in Figure 9 initial ones or equilibrium ones? I cannot understand why the authors adjusted the pH value of ground water.

Minor points

  1. The 7th affiliation was not mentioned in detail.
  2. I am not English speaker, so I should not point out, but English is too poor. E.g. lines 21 and 22; “highly removal” and “the most challenge” should be “high removal” and “the most challenging”, line 23; “reaction of chitosan, urea and amino thiazole derivative” should be “reaction of chitosan with urea and amino thiazole derivative” (I think urea is not correct, but formaldehyde.), line 25; “nitrogen and sulfur content” should be “nitrogen and sulfur contents”, line 26
  3. There are many typos, e.g. ; line 26; “3.5 and 0 mmol g-1” should be “3.5 and 0 mmol g-1”, line 33; 0.2 cannot be head term of the sentence, line 33; “N” is not normal, but “M” , and if the authors use “M”, “M (M = mol/L)” is better.
  4. Please give spaces between value and unit.
  5. Please keep effective digits, e.g. lines 105-106; “6.62 g”, “8.68 g”, and “4 g”.
  6. What is EPI in line 116?
  7. The preparation of MCH was clearly mentioned in the first part of Scheme 1 using different font size of (i) and (ii).
  8. What is “MCHI” in Figure 5?
  9. Eluent should be listed in Table 5, not in the sentences.
  10. “pb” in Figure 9 should be “Pb”.

Author Response

Response to Reviewer#2

Red: Response to reviewer’s comments

Blue: New information/comment added to the manuscript

We would like to thank you for reviewing our manuscript and we are sorry for your recommendation even we think the aim of the work fits the special issue with the same target of polymer journal.

Open Review

English language and style

(x) Extensive editing of English language and style required
( ) Moderate English changes required
( ) English language and style are fine/minor spell check required
( ) I don't feel qualified to judge about the English language and style

Yes

Can be improved

Must be improved

Not applicable

Does the introduction provide sufficient background and include all relevant references?

( )

( )

(x)

( )

Is the research design appropriate?

( )

( )

(x)

( )

Are the methods adequately described?

( )

( )

(x)

( )

Are the results clearly presented?

( )

( )

( )

(x)

Are the conclusions supported by the results?

( )

( )

(x)

( )

Comments and Suggestions for Authors

Comments

 The paper entitled “Grafting of Thiazole Derivative on Chitosan Magnetite Nano-1 particles for Cadmium Removal; Application for Groundwater 2 Treatment” was reviewed. It includes some new findings for preparation of materials and metal adsorption, but there are so many unclear points and typos, so it is not suitable to be published from Polymers.

 Thanks, and sorry to hear that not suitable for polymer journal, we tried to clarify the unclear points that you assigned. As we mentioned above that it is with the same aims of the polymer journal and fit the targets of this especial issue “Polymers in Water Treatment”.

Main points:

  1. The role of magnetite is not clear. Is it just for recovery of adsorbent? So, why did the authors carried out Fe adsorption by using this material?

Uses of Fe as a supporting material of our work as discussed throughout the manuscript is due to

  • Increasing the surface area of the sorbent through fabrication of nano scale particles, which improve the kinetic sorption (few minutes sufficient to complete sorption)
  • Easily collection from the solution after treatments by magnetic bar as shown in the below figure
  • Easily synthesis and available with low cost

 In abstract, the authors mentioned “titration”, but they should mention which kind of titration was carried out.

Thanks for notification it was corrected; It is volumetric titration and as reviewer#1 suggest we add reference for this

  1. The magnetite is located between two hydroxy groups from epichlorohydrin in the first part of Scheme 1. However, it was incorporated in the first step described in 2.2.1. before crosslinking.

Thanks for the comment, in the first step (section 2.2.1) this is to synthesis the magnetite nanoparticles without any organic coating materials, in which the next steps describe the uses of this product (NPs) with organic derivatives (first step with chitosan and the second for the crosslinking by EPI), so not only the hydroxyl groups from EPI that coated the magnetite, but there are also amines and hydroxyls from chitosan particles. The reactive groups coated the magnetite is the simple illustration of the produced material  

  1. Is “HCO” in the second part of Scheme 1 formaldehyde?

Thanks, yes, it is formaldehyde, and it was corrected

  1. It is well-known that aldehyde reacts with amine to form imine. However, in the second part of Scheme 1, formaldehyde crosslinked amine from chitosan and amine from 2-amino-4-thiazoleacetic acid. Is it correct?

That is right the formaldehyde moiety has a function of crosslinker properties in the acidic solution (acetic acid solution here), which crosslink (as a bridge) the amine from the chitosan and amine in the 2-amino-4-thiazoleacetic acid

  1. Related to comment 4, it seems that 2-amino-4-thiazoleacetic acid was directly attached with amino group of chitosan to form ion complex. Carbonyl peak of MCH-ATA in Figure 3 appeared around 1620 cm-1, which can be identified as not COOH but COO-.

Thanks for notification, yes it is already mentioned in the text that is for carbonyl (COO), this is due to the expected tautomerization of the conjugated bonds in the thiazole moieties. In which the peak at 1620 is also shared with C=C of the thiazole ring ( it was added in the table). It is noteworthy that the small intensity peaks at 1728 and 1727 for the MCH-ATA and after recycling for five cycles respectively, are related to the C=O of the carboxylic acid, these two peaks emphasize this tautomerization. It was corrected and the new peak of COOH was added in the table 

  1. Which is the metal adsorption site? Related to comment 5, the authors should prepare the figure of pH dependency for log D vs. pHeqfrom Figure 6. If the authors obtain the slope of 2 for Cd adsorption, the adsorption is dependent on pH. But if not, this curve was caused by hydrolysis of Cd2+ and deprotonation of amino groups from chitosan and 2-amino-4-thiazoleacetic acid, not from dissociated carboxylic acid.

Actually, as mentioned in the text the slop of the plotting log D vs. pHeq is corresponding to +0.46, +0.51, and +0.57 for MCH, MCH-ATA and MCH-ATA respectively, which indicating two protons from matrix (amines as you suggest) used for binding with one Cd2+ ion.

I think the suggestion from the reviewer parallel to what discussed in the manuscript

  1. The authors described selectivity in 3.3. What was derived from? If it was derived from the attached 2-amino-4-thiazoleacetic acid, how about the stability constant of metal ions with 2-amino-4-thiazoleacetic acid? The adsorption reactions must be very complicated and the materials are not pure, so the discussion seems to be non-sense. Another lot adsorbent has another data.

Actually, we do not understand the reviewer’s comment, we check the behavior of sorption in a polymetallic simulated solution to show the sorption efficiencies of each metal combine with other as well as the behavior of the sorbent in the polymetallic solutions.  

  1. How about the attached amount of 2-amino-4-thiazoleacetic acid on chitosan? Is that related to the Cd adsorption capacity?

We adjust the amount of grafted moieties to be not so little to not make sense in the adsorption performances or too high to avoid stearic hindrance and decreasing the sorption. This is derived from the previous studies on the chitosan base polymer with different amines and carboxylic acids.

  1. The authors described two adsorption mechanism. What these mechanisms express for your adsorption?

If the reviewer means the two products of the sorbent tautomerization (because we don’t represent the sorption scheme), in fact the both products are expected to be found. These rearrangements are verifying by the FTIR spectra (the presence of COO- and COOH; OH, and NH). From the data report in the pHpzc , the sorbent is partially protonated which support such intraparticle rearrangement.

  1. Why did authors describe tautomeric effect of thiazole ring? What this effect contribute to?

Thanks for the comment, actually the tautomerization is useful for removing the doubt and confusion that happened from the presence some of unexpected peaks that not found in the original base moiety

  1. Are the pH values mentioned in Figure 9 initial ones or equilibrium ones? I cannot understand why the authors adjusted the pH value of ground water.

 Sorry it is for the intial pH, we adjust the pH of water only for verifying the variation of sorption capacity, which effect on the functional group that oriented to be favorable toward some elements than others at each value. If you think that not make sense, we can remove it and use the original pH of the collected sample

Minor points

  1. The 7thaffiliation was not mentioned in detail.

Thanks, we support it by further information

  1. I am not English speaker, so I should not point out, but English is too poor. E.g. lines 21 and 22; “highly removal” and “the most challenge” should be “high removal” and “the most challenging”, line 23; “reaction of chitosan, urea and amino thiazole derivative” should be “reaction of chitosan with urea and amino thiazole derivative” (I think urea is not correct, but formaldehyde.), line 25; “nitrogen and sulfur content” should be “nitrogen and sulfur contents”, line 26

Thanks for carful revision and fruitful comments, we correct them and revised the whole manuscript, hope that meet your approval

  1. There are many typos, e.g. ; line 26; “3.5 and 0 mmol g-1” should be “3.5 and 0 mmol g-1”, line 33; 0.2 cannot be head term of the sentence, line 33; “N” is not normal, but “M” , and if the authors use “M”, “M (M = mol/L)” is better.

Thanks for comment we already mentioned it by M expressions except in abstract and it was corrected, and check the other errors; thanks again

  1. Please give spaces between value and unit. Please keep effective digits, e.g. lines 105-106; “6.62 g”, “8.68 g”, and “4 g”.

Thanks for notification and advises

  1. What is EPI in line 116?

This is the Epichlorohydrin, and it was mentioned in the material section with the abbreviation.

  1. The preparation of MCH was clearly mentioned in the first part of Scheme 1 using different font size of (i) and (ii).

Thanks, it is only from the expanding of the second part of scheme, now both are the same after rearrangements

  1. What is “MCHI” in Figure 5?

Thanks, it was corrected to MCH

  1. Eluent should be listed in Table 5, not in the sentences.

Thanks, it was added

  1. “pb” in Figure 9 should be “Pb”.

Thanks, it was corrected

Round 2

Reviewer 1 Report

The authors have addressed all reviewer's comments and the revised manuscript is now of good, publishable quality. Authors may still want to add the error bars for each data point in the curves in Figures 6, 7 and 8. Now the average of the 3 values for each data point is plotted and the error bars can be base on these 3 values. Also, English should be checked for the newly added text in the revised manuscript. Otherwise, the rest of the English spelling and grammar corrections are fine.

Author Response

Response to Reviewer #1

Red: Response to the Academic Editor’s comments

Blue: New information/comment added to the manuscript

We would like to thank you for reviewing our manuscript and we are appreciate your efforts and comments.

Open Review

English language and style

( ) Extensive editing of English language and style required
(x) Moderate English changes required
( ) English language and style are fine/minor spell check required
( ) I don't feel qualified to judge about the English language and style

Yes

Can be improved

Must be improved

Not applicable

Does the introduction provide sufficient background and include all relevant references?

(x)

( )

( )

( )

Is the research design appropriate?

(x)

( )

( )

( )

Are the methods adequately described?

(x)

( )

( )

( )

Are the results clearly presented?

(x)

( )

( )

( )

Are the conclusions supported by the results?

(x)

( )

( )

( )

Comments and Suggestions for Authors

We Thanks the reviewer for his global evaluation and positive recommendation, we appreciate this

The authors have addressed all reviewer's comments and the revised manuscript is now of good, publishable quality. Authors may still want to add the error bars for each data point in the curves in Figures 6, 7 and 8. Now the average of the 3 values for each data point is plotted and the error bars can be base on these 3 values.

Thanks for your notification we add the error bars to the figures (pH, uptake kinetics and sorption isotherms) either founds in the core manuscript or in the SI. As you will see the standard deviation is less than the size of symbols if no error bars are seen

Also, English should be checked for the newly added text in the revised manuscript. Otherwise, the rest of the English spelling and grammar corrections are fine.

Thanks, we check the Language and hope it is suited now and meeting your approval

Reviewer 2 Report

The authors' comments are not sufficient.

7. A) which indicating two protons from matrix (amines as you suggest) used for binding with one Cd2+ ion.

Q) So, how was the rest of two charge of Cd2+ ion neutralized? The authors only considered about the binding.

8. A) Actually, we do not understand the reviewer’s comment, we check the behavior of sorption in a polymetallic simulated solution to show the sorption efficiencies of each metal combine with other as well as the behavior of the sorbent in the polymetallic solutions.

Q) If the adsorption site is related to the carboxylate or thiazole component, it is meaningful to compare the stability constants between Cd2+ ion and simple acetic acid or simple thiazole. Of course, it is not easy to collect such data, but it is really disappointed that the authors did not understand what I meant. I am still not clear the component related to adsorption. It is related to the answer to the comment 9.

10. I am sorry to give wrong comments. The mentioned mechanisms were not two, but four.

Author Response

Response to Reviewer #2

Red: Response to the Academic Editor’s comments

Blue: New information/comment added to the manuscript

First, we would like to thank you for your carful revision, evaluation and your valuable comments. We try best answering the comments and made the corrections, we hope it will meet with your approval.

Open Review

English language and style

( ) Extensive editing of English language and style required
(x) Moderate English changes required
( ) English language and style are fine/minor spell check required
( ) I don't feel qualified to judge about the English language and style

Yes

Can be improved

Must be improved

Not applicable

Does the introduction provide sufficient background and include all relevant references?

( )

( )

(x)

( )

Is the research design appropriate?

( )

( )

(x)

( )

Are the methods adequately described?

( )

( )

(x)

( )

Are the results clearly presented?

( )

( )

(x)

( )

Are the conclusions supported by the results?

( )

( )

(x)

( )

Comments and Suggestions for Authors

Thanks for your evaluation of our work, we try to improve the

The authors' comments are not sufficient.

  1. A) which indicating two protons from matrix (amines as you suggest) used for binding with one Cd2+ion.
  2. Q) So, how was the rest of two charge of Cd2+ion neutralized? The authors only considered about the binding.

Thanks for notification, Yes the two protons are from the amine groups, while the negative charge from the carboxylate ions (which is mentioned in the FTIR), the partial negative functional groups and the lone pair of electrons on donner atoms (N, O, S), which found at slightly acidic pH as assigned by pHpzc, assist to neutralize the positive Cd2+ions.

  1. A) Actually, we do not understand the reviewer’s comment, we check the behavior of sorption in a polymetallic simulated solution to show the sorption efficiencies of each metal combine with other as well as the behavior of the sorbent in the polymetallic solutions.
  2. Q) If the adsorption site is related to the carboxylate or thiazole component, it is meaningful to compare the stability constants between Cd2+ion and simple acetic acid or simple thiazole. Of course, it is not easy to collect such data, but it is really disappointed that the authors did not understand what I meant. I am still not clear the component related to adsorption. It is related to the answer to the comment 9.

Thanks for understanding about the complicated of stability constants. The several functional sites on the sorbent (i.e., amine, carboxylic/carboxylate, and hydroxyls) are used in the binding with metal ions. While the mechanism of bending depending on the pH (protonated/ deprotonated functional groups) for possible electrostatic/ non-electrostatic binding or ionic exchange mechanism [1-5].

[1] Tran, H.N. and Chao, H.P., 2018. Adsorption and desorption of potentially toxic metals on modified biosorbents through new green grafting process. Environmental Science and Pollution Research, 25(13), pp.12808-12820.

[2] Khademian, Einallah, Ehsan Salehi, Hamidreza Sanaeepur, Francesco Galiano, and Alberto Figoli. "A systematic review on carbohydrate biopolymers for adsorptive remediation of copper ions from aqueous environments-part A: Classification and modification strategies." Science of the Total Environment 738 (2020): 139829.

 [3]Zhang, M., Yang, P., Lan, G., Liu, Y., Cai, Q. and Xi, J., 2020. High crosslinked sodium carboxyl methylstarch-g-poly (acrylic acid-co-acrylamide) resin for heavy metal adsorption: Its characteristics and mechanisms. Environmental Science and Pollution Research, 27(31), pp.38617-38630.

[4] Suručić, L.T., Janjić, G.V., Rakić, A.A., Nastasović, A.B., Popović, A.R., Milčić, M.K. and Onjia, A.E., 2019. Theoretical modeling of sorption of metal ions on amino-functionalized macroporous copolymer in aqueous solution. Journal of Molecular Modeling, 25(6), pp.1-12.

[5] Kilislioglu, Ayben, ed. Ion exchange technologies. BoD–Books on Demand, 2012.

  1. I am sorry to give wrong comments. The mentioned mechanisms were not two, but four.

 As mentioned, the behavior of sorption depending on the pH of the solution and as the electron pairs are available at these pH values it is easily to make the tautomerizations and the four tautomer expected to be found [1,2]

[1] Metzger, J. V. "Thiazoles and their benzo derivatives." (1984): 235-331.

[2] Kusakiewicz-Dawid, A., Porada, M., Dziuk, B. and Siodłak, D., 2019. Annular Tautomerism of 3 (5)-Disubstituted-1H-pyrazoles with Ester and Amide Groups. Molecules, 24(14), p.2632.

Round 3

Reviewer 2 Report

The revised article entitled “Grafting of Thiazole Derivative on Chitosan Magnetite Nano-1 particles for Cadmium Removal; Application for Groundwater 2 Treatment” was re-reviewed. The authors successfully answered to the reviewers’ comments and revised the manuscript. So, it can be published from Polymers.

Author Response

Thanks for the globally recommendation, we appreciate your efforts